# Common Errors in the Management of Idiopathic Clubfeet Using the Ponseti Method: A Review of the Literature

**DOI:** 10.3390/children10010152

**Published:** 2023-01-12

**Authors:** Sean B. Youn, Ashish S. Ranade, Anil Agarwal, Mohan V. Belthur

**Affiliations:** 1College of Medicine, University of Arizona–Phoenix, 475 N 5th St., Phoenix, AZ 85004, USA; 2Blooming Buds Centre for Pediatric Orthopaedics, Deenanath Mangeshkar Hospital, Pune 411004, India; 3Department of Pediatric Orthopaedics, Chacha Nehru Bal Chikitsalaya, New Delhi 110031, India; 4Department of Orthopedics, Main Building, Clinic B, Phoenix Children’s Hospital, 1919 E Thomas Rd, Phoenix, AZ 85016, USA

**Keywords:** congenital idiopathic clubfoot, Ponseti method, common errors, casting, bracing, tenotomy, pitfalls

## Abstract

Congenital talipes equinovarus is one of the most prevalent birth defects, affecting approximately 0.6 to 1.5 children per 1000 live births. Currently, the Ponseti method is the gold-standard treatment for idiopathic clubfeet, with good results reported globally. This literature review focuses on common errors encountered during different stages of the management of idiopathic clubfeet, namely diagnosis, manipulation, serial casting, Achilles tenotomy, and bracing. The purpose is to update clinicians and provide broad guidelines that can be followed to avoid and manage these errors to optimize short- and long-term outcomes of treatment of idiopathic clubfeet using the Ponseti method. A literature search was performed using the following keywords: “Idiopathic Clubfoot” (All Fields) AND “Management” OR “Outcomes” (All Fields). Databases searched included *PubMed*, *EMBASE*, *Cochrane Library*, *Google Scholar*, and *SCOPUS* (age range: 0–12 months). A full-text review of these articles was then performed looking for “complications” or “errors” reported during the treatment process. A total of 61 articles were included in the final review: 28 from *PubMed*, 8 from *EMBASE*, 17 from *Google Scholar*, 2 from *Cochrane Library*, and 6 from *SCOPUS*. We then grouped the errors encountered during the treatment process under the different stages of the treatment protocol (diagnosis, manipulation and casting, tenotomy, and bracing) to facilitate discussion and highlight solutions. While the Ponseti method is currently the gold standard in clubfoot treatment, its precise and intensive nature can present clinicians, health care providers, and patients with potential problems if proper diligence and attention to detail is lacking. The purpose of this paper is to highlight common mistakes made throughout the Ponseti treatment protocol from diagnosis to bracing to optimize care for these patients.

## 1. Introduction

Congenital talipes equinovarus is one of the most common birth defects affecting the lower extremity. The components of a true idiopathic clubfoot include midfoot cavus, forefoot adduction, hindfoot varus, and ankle equinus (CAVE). The global prevalence of clubfoot is between 0.6 and 1.5 per 1000 live births, and approximately 80% of the cases are from low-to-mid-income countries (LMIC). One study showed that amongst WHO-designated LMIC, the prevalence was between 0.51 and 2.03 per 1000 live births: 0.51 in China, 1.11 in Africa, 1.21 in Southeast Asia, 1.74 in the Americas, and 2.03 in Turkey [1]. Recent evidence suggests that the etiology of idiopathic clubfoot is multifactorial including both genetic and environmental factors [2,3,4]. While many treatment protocols have been used in the past, the Ponseti method is currently the gold-standard treatment method in the management of idiopathic clubfeet. Dr. Ignacio Ponseti developed this treatment protocol in 1948 and published his first clinical outcome study using this protocol in his groundbreaking paper in 1963 [2,4]. The Ponseti method utilizes the viscoelastic properties of the foot and the kinematic coupling of the subtalar joint motion to achieve correction of the clubfoot [2]. This treatment protocol, separated into casting and bracing phases, has been adopted around the world due to its simplicity, reliability, minimal invasiveness, and high success rate both in the short and long term [2]. The casting phase consists of gentle manipulation followed by weekly serial stretch casting of the clubfoot using long-leg Ponseti casts to correct midfoot cavus, forefoot adduction, hindfoot varus, and ankle equinus (CAVE) in that specific order. This is followed by a percutaneous tendoachilles tenotomy in most cases (about 90%) to achieve ankle dorsiflexion of more than 10 degrees and a final long-leg Ponseti cast with the foot abducted to about 60 degrees for 3 weeks. The bracing phase consists of using bilateral boots and a foot abduction bar full time for 3 months and part time (night and nap time) for up to 4–5 years of age to maintain correction and prevent relapse of the clubfoot deformity. There have been a multitude of studies showing its effectiveness in both developed and developing LMIC countries, with short-term success rates of approximately 90% and long-term rates at 80% [5,6,7,8,9,10,11,12,13,14]. Despite this success, errors can be made during the treatment process, resulting in complications if there is not close attention to detail due to the intensive nature of this treatment program [15,16]. Ponseti highlighted these errors and recommended steps to avoid them in his classic paper in 1997 [15]. With the Ponseti method being applied globally, we decided to revisit the common errors in the treatment of idiopathic clubfoot and update the information about it using an extensive literature search to create this comprehensive review paper. The purpose of this review is to outline common errors encountered during the different stages of management of the idiopathic clubfoot using the Ponseti method by healthcare professionals and to discuss solutions to help avoid and rectify them to optimize the outcomes of the Ponseti method.

## 2. Materials and Methods

This study was IRB-exempt, as it did not directly involve human subjects (secondary research study). A literature search was performed using the following keywords: “Idiopathic Clubfoot” (All Fields) AND “Errors” OR “Management” OR “Outcomes” (All Fields). Databases searched included *PubMed*, *EMBASE*, *Cochrane Library*, *Google Scholar*, and *SCOPUS* (age range: 0–12 months). Duplicate articles were excluded. This returned a total of 183 articles. A full-text review of these articles was then performed looking for “complications” or “errors” reported during the treatment process using the Ponseti method. The bibliography (references) of each article was hand-searched to look for other relevant articles reporting errors associated with the treatment of idiopathic clubfoot. English-language articles published between January 1963 and March 2022 regarding management of idiopathic clubfeet were included in this review. Articles regarding non-idiopathic clubfoot, syndromic clubfoot, treatment method other than the Ponseti method, and case reports were excluded. This was a review of the literature and not a systematic review, and hence, PRISMA guidelines were not utilized.

## 3. Results

A total of 61 articles were included in the final review: 28 from *PubMed*, 8 from *EMBASE*, 17 from *Google Scholar*, 2 from *Cochrane Library*, and 6 from *SCOPUS*. There were a total of six articles that elaborated on the errors and complications associated with the Ponseti method while managing idiopathic clubfeet. Errors encountered in the management of idiopathic clubfoot using the Ponseti method were grouped based on the stage of treatment into the following categories (Table 1 and Table 2): (1)Errors in diagnosis.(2)Errors in manipulation.(3)Errors during serial casting.(4)Errors during percutaneous Achilles tenotomy.(5)Errors during bracing phase.
children-10-00152-t001_Table 1Table 1Most common errors in clubfoot management based on stage of treatment using the Ponseti method.Stages in ManagementTiming/ManeuverProblem/ErrorSolution**Diagnosis**
Errors in diagnosisUnderstand differences between the four types of clubfoot **Manipulation**Set-upCompetent/trained applicator with understanding of Ponseti biomechanicsTime spent for manipulation Cavus correctionHindfoot varus and forefoot adduction correctionFoot abductionAnkle equinus correctionHigh-stress environment for child and health care teamInadequate correctionPronating footExternal rotation of the foot prior to calcaneus varus correctionKite’s error: pressure applied over calcaneocuboid jointFailure to fully abductDorsiflexing foot prior to correction of foot supination and heel varusQuiet room, distractions for child (music, food, parent’s voice).Spend at least a minute before casting Supinate forefoot Avoid doing thisUsing the lateral talar head as the fulcrumAbduct the foot to 60 degrees The correction follows the order CAVE**Serial casting**Cast applicationUse of excess paddingCast material and molding Excess pressure when castingEarly removalCare of castLess than two people helping with applicationSlippage of the foot and worsening of the deformityFailure to mold can cause cast slippageUse of below-knee castsMoisture lesions, skin macerations, dermatitis, pressure soresLoss of correctionCast slippageTwo-person team for castingApply a well-contoured cast with minimal padding except at the toes and the groin.Mold well; use rapid-setting cast materialMold wellUse of toe-to-groin castsGentle pressure and move fingers around while castingRemove just before next castParental education **Tenotomy**Excess local anesthesiaProcedure too distalIncomplete procedureNeurovascular structure injuryTissue swelling Incomplete tenotomyRelapse of deformityPotentially dangerous, permanent complicationsUse topical anesthetic cream and inject local after the procedureTenotomy 1.5 cm proximal to os calcisLook for improved ankle dorsiflexion to 15 degrees and listen for a “pop” soundPerform tenotomy medial to lateralMonitor patient post tenotomy for soakage in cast**Bracing and follow-up**Failure to comply with use instructions and irregular follow-upToo tight a braceToo loose a brace CommunicationBrace designEvaluate social determinants of health (SDOH), parent loadRelapse of deformityPressure area over anterior anklePressure area over the heelPoor provider–family communication Poor brace designNot evaluating SDOHParental education Appropriate brace tighteningAppropriate brace tighteningTransform care usingrelationship-based communication skillsBetter brace designEvaluate SDOH and parental load to predict non-compliance
children-10-00152-t002_Table 2Table 2Errors during the Ponseti method reported in the literature.Author/Type of StudyYearPublishedErrors in ManipulationErrors in CastingErrors in TenotomyErrors in BracingPonseti IV [15]Review1997Kite’s errorForefoot pronation or eversionExternal rotation of foot while calcaneus is in varusEquinus correction before achieving midfoot inversion and heel varusManipulation without immobilizationUse of below-knee casts
Failure to use splints that fix foot in external rotation for 3 months full time and at night for 2–4 years Baindurashvili [17]Case series2012
Skin irritationMacerationPressure ulcers

Badoo AR [18]Case series2016Failure to localize talar head prior to treatmentEquinus correction before achieving midfoot inversion and heel varusCast slippageUse of below-knee castsSpoilage of casts with urine or fecesExcessive pressure when casting

Agarwal A [19]Case series2016
Moisture lesions, hematomas, dermatitis, pressure sores, fracturesExcessive bleedingDermatitis due to occlusion, pressure sores, tenderness at tenotomy siteChotigavanichaya C [20]Case series2016
Cast loosening or breakage with resulting wound dehiscenceCast skin irritationPressure soresInfectionSuperficial wound infectionsWound bleedingInadequate foot-forming braces leading to relapse and necessitating secondary surgeryNoncompliance due to discomfort of brace and lack of parental educationRaykov D [9]Review2017Kite’s errorExternal rotation to correct foot adductionFoot pronationEquinus correction before achieving midfoot inversion and heel varusFailure to fully abduct footInadequate cast fittingUse of below-knee casts
Early removal of FAB

## 4. Discussion

Although the Ponseti technique has useful applicability, it is a methodical procedure, requires close attention to detail, and has a learning curve [21]. Its precise technical details make it prone to error/complications, especially for the early learners. Ponseti recognized this fact and highlighted some of the common errors and steps to avoid them in his classic paper in 1997 [15]. Lehman et al. noted a complication rate of 10.2%, citing problems such as cast intolerance, skin maceration, and abrasions [20]. Dobbs et al. and Changulani et al. discussed bleeding complications during the tenotomy procedure [15,22]. Raykov focused on manipulation errors, including incorrect timing of equinus correction and external rotation of the foot to correct foot abduction [15]. In addition, there are also significant variations with how it is implemented and who performs the treatment, i.e., physical therapists, nurses, orthopedic surgeons, podiatrists, etc. Zhao et al. discussed deviations from the original Ponseti method, noting four studies that used completely different bracing designs, three that had conflicting bracing protocols, and eight that had different criteria for bracing compliance [23].

1.
**Errors in diagnosis:**


There are four types of clubfoot: idiopathic, neuromuscular, syndromic, and positional. Because some clubfeet require specific deviations from the original Ponseti method to adequately treat the foot abnormality, it is vital that physicians are diligent during the initial physical examination to ensure the correct diagnosis is made.

Idiopathic clubfoot is the most common type and is further classified into typical (TIC) and atypical (ATIC) types, both presenting with isolated foot abnormalities and varying degrees of severity. In the TIC, there is a greater amount of fibrotic tissue in the gastrocsoleus, tibialis posterior and long toe flexor muscles, the posterior ankle, and medial tarsal ligaments. In the ATIC, the gastrocsoleus and plantar intrinsic muscles are more involved. This results in a shorter and stubbier foot, severe plantar flexion of all the metatarsals, a deep transverse crease in the sole, a deep crease above the heel, and hyperextended hallux. While TIC is treated with the Ponseti method, modifications are made with ATIC including use of a supination maneuver, midfoot dorsiflexion technique, early tenotomy, and placement of the foot abduction orthosis at less external rotation (40 degrees) [23,24,25].

Neuromuscular clubfoot (NMC) is secondary to disorders such as spina bifida, congenital myotonic dystrophy, congenital myopathies, or congenital peroneal neuropathy [26]. Patients with NMC may present with a drop-toe sign, described as resting toes in a plantar flexion position without toe extension following plantar stimulation. This contrasts with idiopathic clubfeet that show toe dorsiflexion while moving the foot and active toe and ankle dorsiflexion upon plantar stimulation [27]. While treatment protocols are like those for TIC, the cause of relapse is different. In a case series by Lovell et al., 6% of their patient population had relapse events after 7 years of age despite most relapses occurring before the age of 6 years. Thirty-one percent of these patients had a late neuromuscular condition diagnosis such as Charcot-Marie-Tooth type 1A or myasthenia gravis [28]. Therefore, it is important for physicians to realize that relapses may not be due to poor treatment protocol but from an underlying previously undiagnosed neuromuscular condition.

Syndromic clubfoot is usually secondary to other medical conditions such as arthrogryposis, tibial hemimelia, fibular hemimelia, diastrophic dysplasia, and constriction band syndrome, among others. Studies comparing idiopathic and syndromic clubfoot have reported the need for more serial casts to correct the foot deformity (mean of 6.5 versus 4 in the TIC), a greater need for surgical intervention, and a higher rate of relapse for the latter [29,30].

Positional clubfoot presents as a flexible, typically fully developed foot that is inverted and internally rotated due to prolonged positioning in utero. This is commonly associated with larger babies, oligohydramnios, and multiple pregnancies. There are no differences in the length and width of the foot, and the foot has full range of motion in all joints. Because there are no bony or soft tissue abnormalities, conservative treatment of stretching and range of motion exercises are typically sufficient in correcting the clubfoot [31,32].

Appropriately grading the severity of the clubfoot deformity after precise diagnosis using validated measures before and during the treatment process is important to guide the treatment process and monitor the progress of the treatment protocol [5,33].

2.
**Errors in manipulation:**


To preface, both the manipulation and serial casting process should be performed in a low-stress environment: a quiet room with the child lying supine at the end of the table with at least two people present (a holder and a casting person) [33,34]. (Figure 1)

Many measures have been suggested to soothe the child and make the casting process pleasant for both providers and receivers. The child can be appropriately distracted with music or sugar solution (it is advisable to keep the child hungry for about 2 h prior to the appointment), and the parent may whisper to the child during casting to optimize the casting experience [17,18,19].

The first step in the manipulation and serial casting process is to correct midfoot cavus (high-arched foot) by elevating the first metatarsal to correct forefoot pronation [2,15,25,35]. While there is excessive supination of the foot, the forefoot is pronated in relation to the hindfoot, and thus, one common error is believing that pronating the forefoot will correct the deformity. This, however, increases the midfoot cavus and does not abduct the adducted and inverted calcaneus, which remains locked under the talus [30]. In turn, a new eversion deformity is created through the mid- and forefoot, leading to a bean-shaped foot [5,15,33].

The second step is correcting hindfoot varus and forefoot adduction by maximally abducting the supinated foot with counter pressure against the lateral aspect of the head of the talus [2,15,25,36]. (Figure 2)

With casting, the calcaneus starts to evert and abduct under the talus as the hindfoot transitions from varus to valgus. Physicians must be aware not to externally rotate the foot to fix abduction while the calcaneus is still in varus, as this displaces the lateral malleolus posteriorly by externally rotating the talus in the ankle mortise. When abducting the foot at the midtarsal joints, if pressure is applied over the fibula or calcaneocuboid joint, the kinematic coupling of the subtalar joint would be blocked and prevent correction of the heel varus deformity; this is known as Kite’s error [30]. In addition, holding the leg at the ankle or above while abducting the foot puts a rotational force on the talus and causes the fibula to be forced backwards [33]. Finally, failure to fully abduct the midfoot will result in insufficient reduction of the head of the talus, heel valgus, and subtalar dorsiflexion. This also highlights the importance of identifying the lateral talar head because the talar head acts as a fulcrum point under which the whole foot (calcaneopedal block) is abducted. In one study, the talar head was difficult to locate due to abnormal formation of feet in 20 of 123 clubfeet (16.3%), complicating treatment [2,15,25,37]. The last step is to attempt correction of the ankle equinus, which is caused by an abnormally short Achilles tendon [2,15,25]. Pushing the foot into dorsiflexion before correcting forefoot adduction and heel varus can result in a midfoot break and a rocker-bottom foot. This can also create a flat-top talus due to pressure on the talar head. To avoid these errors, the physician must ensure that there is more than 60 degrees of abduction of the foot in relationship to the anterior aspect of tibia and that the heel is either in neutral or valgus prior to dorsiflexing the talar head [33].

3.
**Errors in casting:**


Serial stretch casting is a vital component of the Ponseti treatment method, and errors can lead to relapses and redevelopment of foot deformities. First, the cast should not be placed by one person; instead, one individual (leg holder) should ensure the foot and leg are being held in the corrected position while another applies the cast, with a parent ideally (sitting on the opposite side of the leg being casted) there to calm the child [17,18,19,33]. Careful consideration of the type of material used for casting is important. Webril 100% cotton cast padding is commonly used, as it is user-friendly, pliable, not constricting, and nicely conforming. Rapid-setting plaster of Paris is generally used as the casting material because of its easy availability, molding characteristics, and low cost. The cast should also be positioned correctly with appropriate cast padding around the toes, one layer in the middle with 50% overlap, and appropriate thickness around the groin [33]. Correct sequencing of the cast application process decreases the chance of complications. The below-knee part of the cast should be applied first, allowed to set, and then should be followed by the above-knee part [2,25,33]. If the entire leg cast is put on at one time, there may be areas of the cast that are too tight and others that are loose. Physicians must also avoid using too much padding because it can lead to slipping or migration of the child’s foot in the cast, resulting in soft tissue damage, a smashed foot, and worsening of the deformity. Toe-to-groin casts are recommended instead of below-knee casts because the latter is unable to prevent ankle and talus rotation and, in turn, holds the foot in abduction under the talus to [2,25,30,33]. (Figure 3)

One study showed that 10 of 123 clubfeet slipped out of the cast, with four of them due to the lack of a toe-to-groin cast [37].

Applying excess pressure during casting may cause tissue edema. Excessive sweating can lead to moisture lesions, skin macerations, or dermatitis [38]. In addition, unrelieved pressure over a bony prominence can cause pressure sores, which can progress to tissue ischemia and necrosis. In one study, out of 180 children treated with the Ponseti method, 19.7% had soft tissue lesions, the majority of which presented with superficial lesions [39]. Skin macerations, though, were more common than soft tissue lesions. For these reasons, it is important for parents to monitor the child for cast slippage, excessive crying or skin discoloration. The physician should take the cast off immediately to inspect the leg and treat skin lesions appropriately with antiseptic dressing and re-cast after the problem resolves.

Cast application with excessive force may lead to fractures. In a study by Ranjan et al., distal tibia and fibula fractures were noted. The risk factors were neglected, syndromic feet, feet requiring more than 10 casts, post-tenotomy dorsiflexion less than 10 degrees, and casting by physician with less than 3 years of casting experience. Osteopenia resulting from disuse and prolonged casting may additionally contribute. Forceful dorsiflexion was the common cause of fracture, and most healed without consequences (Figure 4, Figure 5, Figure 6, Figure 7 and Figure 8) [40].

Early removal of the cast can result in loss of correction; thus, parents must be informed that casts must be worn till the next casting appointment to maintain those corrections [17,33]. Casts should only be removed in the clinic rather than at home because it is helpful to see the position of the foot in the previous cast, and casts removed more than an hour prior to the new cast can result in loss of correction. The cast should not be removed using a cast saw and rather should be removed manually after wetting and loosening the cast on the day of the next appointment; this is because there is not enough padding under the cast, and the infant’s delicate skin can easily be injured by the saw [33,41].

Other errors include not scoring the deformity before applying the next cast or keeping the serial record using validated clubfoot scores [5,13,33]. Failure to do this diligently results in missing or unsatisfactory results. Parents may not be able to recognize cast slippage or softening if inadequately educated [5,13,33]. Previous clinical and simulation experience (of >10 casts) related to the Ponseti method has been shown to correlate with improved performance [21,42,43].

4.
**Errors during percutaneous Achilles tenotomy:**


There is residual ankle equinus contracture following correction of other deformity components in most children with idiopathic clubfeet (about 90%) following the serial casting process. If there is less than 10 degrees of ankle dorsiflexion, then an Achilles tenotomy is recommended to prevent relapse of clubfoot deformity and better brace tolerance [2,5,15,33]. A small, percutaneous incision is made over the posteromedial aspect of the Achilles tendon about 1.5–2 cm proximal to the os calcis tuberosity to sever the tendon, and the foot is then placed into a long leg cast with the foot abducted greater than 60 degrees in relation to the tibia, where the tendon regrows in a lengthened position to allow for greater ankle dorsiflexion. Common errors of this procedure include (1) injecting too much local anesthesia, which leads to tissue swelling and complicates identification of the Achilles tendon, and (2) performing a tenotomy that is less than 1.5 cm from the calcaneal tuberosity, thus leading to an incomplete tenotomy [33]. To avoid this, look for improved ankle dorsiflexion to >15 degrees, listen for a “pop” sound, and feel with a hemostat before and after for completion. Finally, to avoid neurovascular structure injury, start the procedure from the unsafe side (medial: tibialis posterior tendon, flexor digitorum longus tendon, posterior tibial artery, tibial nerve, and flexor hallucis longus tendon) and move towards the safe side (lateral: lesser saphenous vein, sural nerve, and peroneal nerve) [44]. Due to the percutaneous nature of the procedure, excessive bleeding from the posterior aspect of the ankle is likely due to poor technique or anomalous vessels [45,46]. A mini-open technique of percutaneous Achilles tenotomy may be safer [46]. Some authors argue that performing the surgery under sedation or general anesthesia may be safer than under local anesthesia [47].

5.
**Errors during the bracing phase:**


After obtaining correction of all components of the clubfoot deformity through serial casting and tenotomy, patients must wear a foot abduction brace (FAB) full-time for the first 3 months and part-time (night and nap time) for 4–5 years to maintain the correction and prevent relapse [48,49]. Ninety percent of relapses happen before the age of 5 years, which is why currently, the recommendation is to continue wearing the brace until that age even if the parents believe the deformity appears corrected prior. Noncompliance or failure to wear the FAB is one of if not the most significant factor leading to relapse of the deformity [36]. Two studies showed brace noncompliance rates of 41% and 36% were 183 and 120 times more likely to relapse compared to children who fully followed their bracing protocols [36,50]. Another error in bracing is attempting to fit a brace before the foot is fully corrected. Agarwal et al. showed that 7–10% of patients who had not achieved greater than 15 degrees of dorsiflexion, required stretch casts after tenotomy before bracing could be started [51]. One of the largest areas of variability in the Ponseti method is brace design, and as such, poor brace design will increase the risk of brace intolerance, as the brace may not be comfortable to wear and results in non-compliance with the bracing protocol [16,38]. Use of the correct type of brace is important because the child will only wear the brace if it is comfortable [48,52,53]. (Figure 9 and Figure 10)

Applying the FAB brace too tightly can cause an anterior ankle pressure area, while applying too loosely can cause foot slipping out of the brace a posterior heel pressure area, leading to brace intolerance (Figure 11 and Figure 12).

Awareness of possible neurodevelopmental challenges in children with idiopathic clubfoot is important as it determines the type, tolerability, and effect of bracing on the clubfoot treatment. Neurodevelopmental disorders may also be responsible for sleep disturbances rather than braces. A broader intervention approach beyond the traditional clubfoot treatment, including speech and language therapy, motor training, behavioral interventions, and/or psychopharmacology, may be needed on a case-to-case basis [54]. Overcorrected clubfeet resulting in a flatfoot is a recognized complication following treatment of the idiopathic clubfoot in both the affected foot as well as the contralateral foot (secondary to bracing) [55,56]. The bracing protocol may need to be modified to address this issue.

Social determinants of health (SDOH) must be considered when using the Ponseti method, especially while treating patients in resource-constrained and disadvantaged areas. Limited resource availability, medical costs, and travel are some of the many factors that need to be analyzed closely [57]. A study in Uganda showed that despite manipulation and treatment being free of charge, weekly transport costs became a significant hindrance, with about 87.7% of the population living in rural areas and with very few having their own vehicles [58]. In a low-resource clinic in Haiti, patients experienced longer manipulation and casting phases for an average of 5.4 weeks than those in high-resource settings; additionally, tenotomy was performed in 31% of patients compared to 81% in more developed countries, leading to a 19% higher relapse rate [59]. Furthermore, braces are required for multiple years to maintain the correction, and with the growth rate of children, multiple braces are needed, increasing the total cost of treatment.

During clubfoot treatment, braces are required for several years post deformity correction. The multiple braces sizes required to complete the prolonged brace-wear period drastically increases the overall cost of clubfoot treatment. This can lead caregivers to try alternative methods and discontinue treatment with the hope that the corrected feet will remain as such. Errors in communication account for about 15% of bracing intolerance [60,61,62]. It is very important to use relationship-based communication skills to engage and spend time with the parents, establish an empathetic relationship with them to modify their behavior, educate them, enlist their help, and make them partners in treatment of their child [63,64,65]. It has been shown that 3D-printed models may help to improve physician–patient communication [66]. However, these are expensive, and it may not be easily available for facilities to develop these models in low-to-middle-income countries. Identifying families at risk of dropping out from clubfoot care enables support to be instigated. The parent load indicator, in parallel with the initial clubfoot severity assessment, may help clinicians better appreciate the demand that treatment will place on parents, the associated risk of drop-out, and the opportunity to enlist support [64,65].

Although the Ponseti method is gaining more traction across the world, a lack of physicians, especially those who are well-trained in this treatment method, remains a major problem in LMIC. Those physicians who are trained are also tasked with not only treating patients with other diseases but also training other clinicians. One study in Tanzania showed that a self-guided e-learning course on the Ponseti method along with a hands-on skills training day significantly improved test scores regardless of prior educational background [67]. Providing LMIC with simple strategies such as this will greatly improve care for patients in those regions [5,21,33,42,43,67].

The Ponseti method is based on a sound knowledge of foot biomechanics (kinematic coupling of the subtalar joint), and failure to recognize this increases the chances of errors during its application [5,15,33]. These errors/complications often threaten progress made to correct the deformities and ultimately may leave the child with an inadequately treated clubfoot. This highlights the importance of a meticulous treatment plan with great attention to details and parental education on the casting and bracing process. While the Ponseti method is already a proven way to treat congenital talipes equinovarus, learning from errors, avoiding them, and tackling them when discovered, is the way forward to success and a deformity-free child.

Our study has several limitations. It is a review of literature, and there are limited randomized controlled trials on the subject. There is heterogeneity of patient/physician-reported outcomes. We acknowledge that this may result in inaccurate reporting of errors. Furthermore, there are no data on the level of competency/experience of the healthcare personnel performing the Ponseti casting. We do not know with certainty the optimum experience to minimize errors. There is a need for universal error/adverse outcome reporting system while treating clubfeet. We have added this information to the manuscript.

## 5. Conclusions

While the Ponseti method is currently the gold standard in clubfoot treatment, its precise and intensive nature can present clinicians, health care providers, and patients with potential problems if proper diligence is not used. These problems are further magnified in developing countries where transportation, lack of resources and physicians, lack of training, lack of education or illiteracy of parents, inadequate adherence to protocols, and medical costs are also significant barriers to overcome. The purpose of this paper is to highlight common mistakes made throughout the Ponseti treatment protocol from diagnosis to bracing in order to optimize care for these patients.

## Figures and Tables

**Figure 1 children-10-00152-f001:**
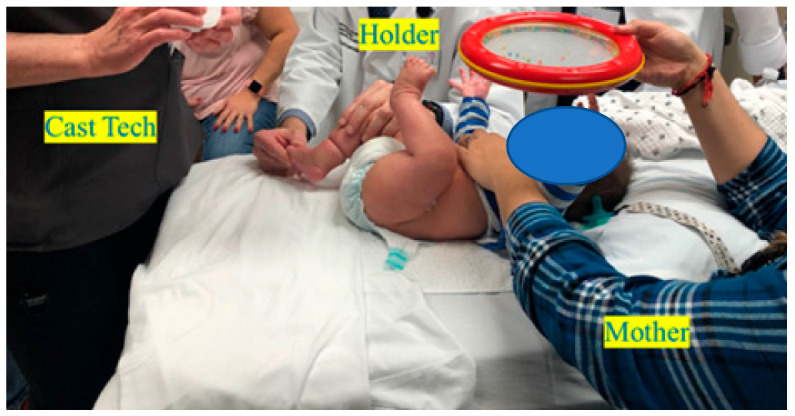
Correct set up for the Ponseti serial casting process.

**Figure 2 children-10-00152-f002:**
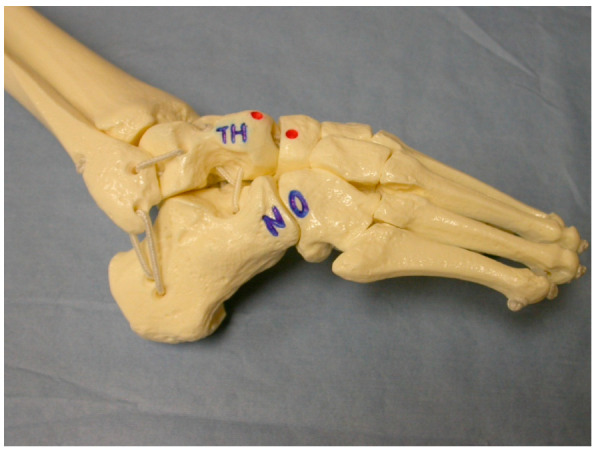
Ponseti foot model showing where to apply pressure on the lateral talar head (TH) and which area (calcaneocuboid joint) to be avoided (NO).

**Figure 3 children-10-00152-f003:**
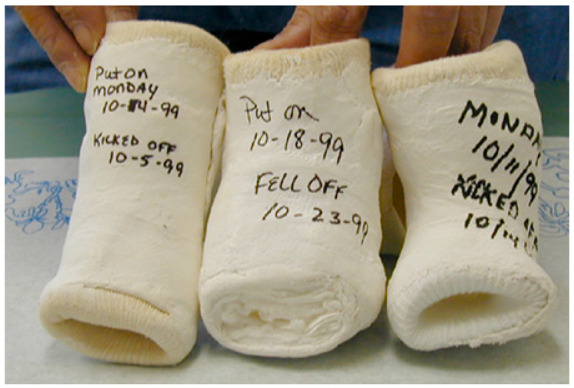
Clinical picture showing below knee casts with excessive padding resulting in cast slippage (to be avoided).

**Figure 4 children-10-00152-f004:**
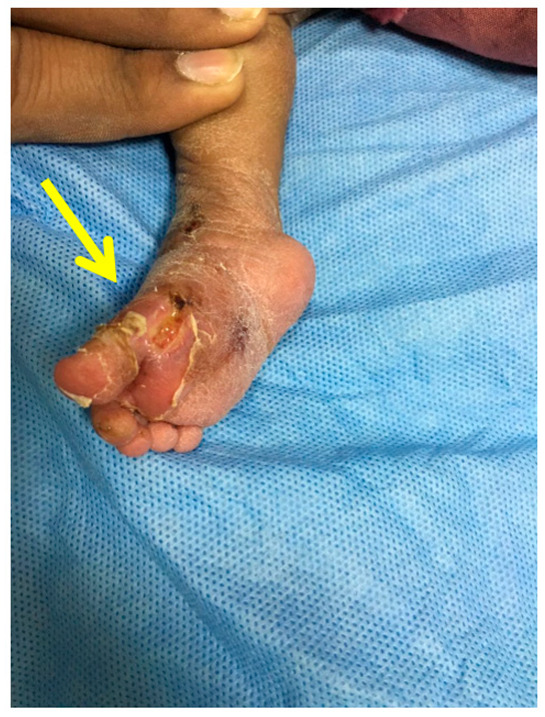
Clinical photograph showing a pressure sore over the medial side of the foot (pointed by the arrow) due to inadequate padding.

**Figure 5 children-10-00152-f005:**
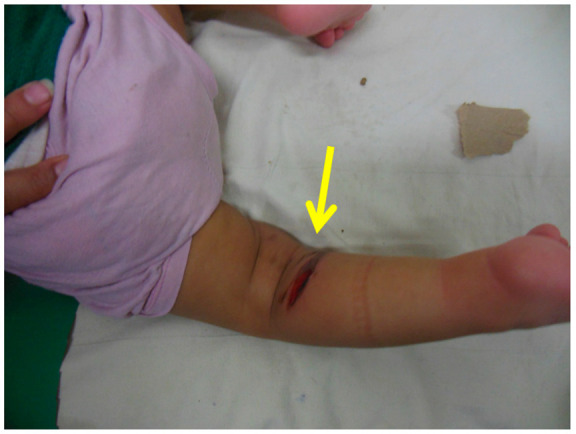
Clinical photograph showing a pressure sore (pointed by the arrow) on the posterior aspect of the proximal calf due to pressure from the plaster wrinkle.

**Figure 6 children-10-00152-f006:**
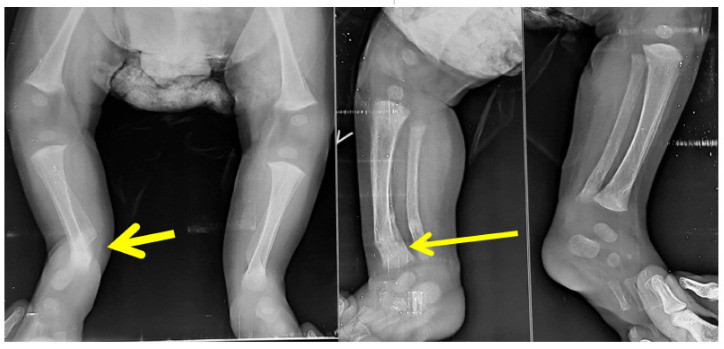
Radiographs showing fracture of the distal tibia metaphysis (pointed by the arrow) during Ponseti serial casting.

**Figure 7 children-10-00152-f007:**
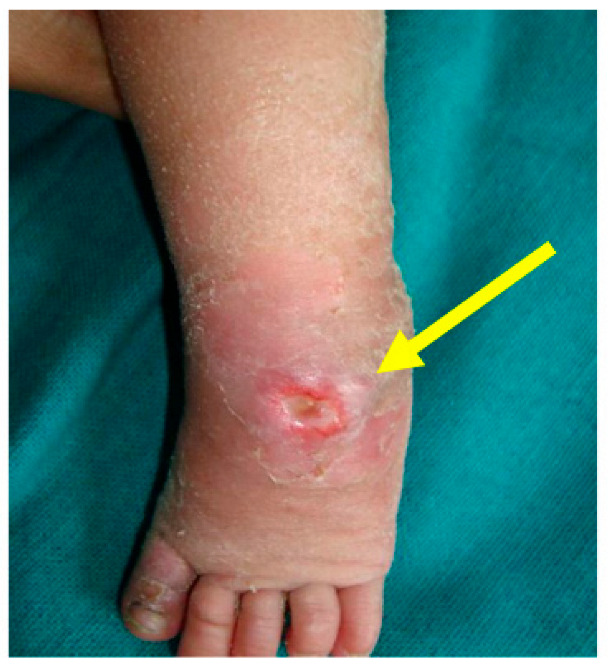
Clinical photograph showing a pressure area over the lateral talar head (pointed by the arrow) due to excessive pressure in one area during the casting process.

**Figure 8 children-10-00152-f008:**
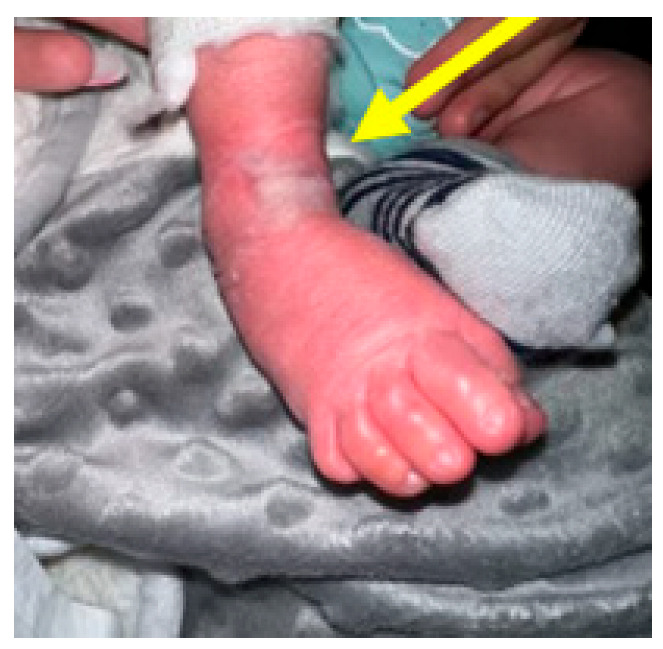
Clinical photograph showing an area of skin necrosis over the anteromedial aspect of the right ankle (pointed by the arrow) with swelling of the toes due to a cast that was too tight.

**Figure 9 children-10-00152-f009:**
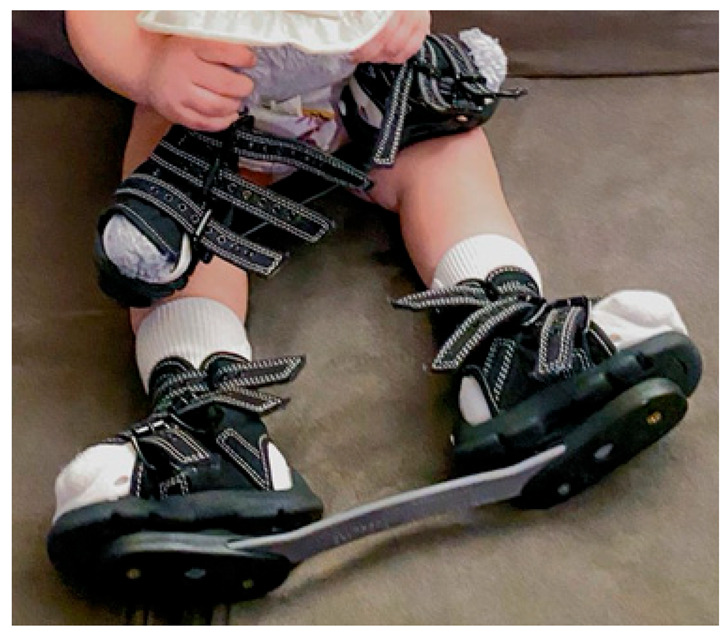
Clinical picture showing a child with fully corrected B/L clubfeet in a correctly designed foot abduction brace.

**Figure 10 children-10-00152-f010:**
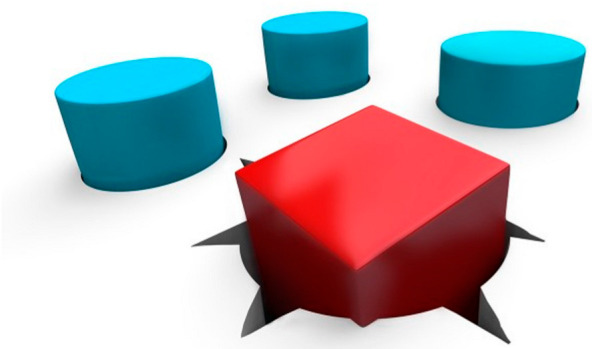
Picture showing the principle of trying to fit a square peg into a round hole; this applies to fitting a clubfoot that is not fully corrected into a foot abduction brace (to be avoided).

**Figure 11 children-10-00152-f011:**
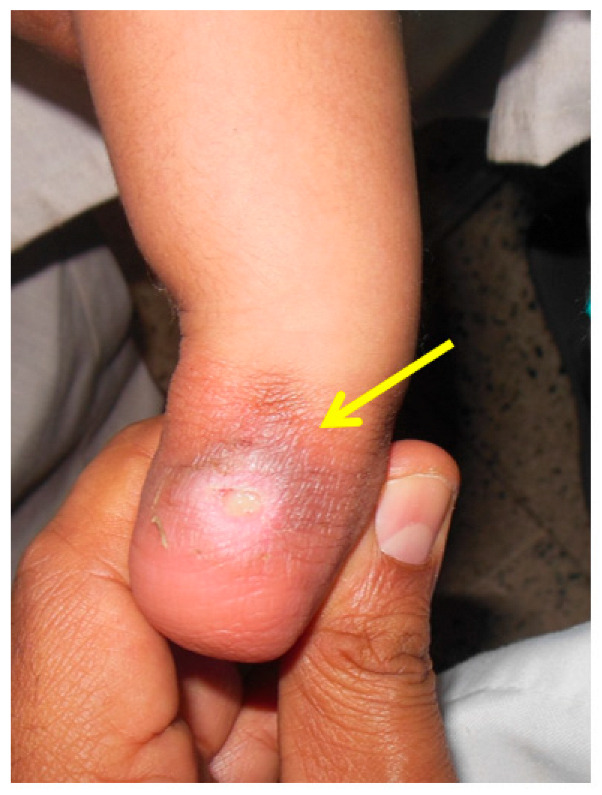
Friction sore on the posterior aspect of the heel (pointed by the arrow) due to the brace being too loose (inadequately tightened).

**Figure 12 children-10-00152-f012:**
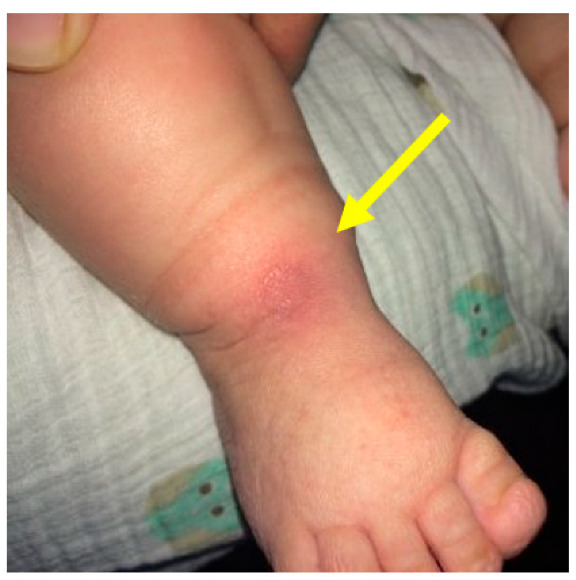
Clinical picture showing an anterior ankle pressure area (pointed by the arrow) caused by a brace tightened excessively.

## Data Availability

Not applicable.

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
