# Peer review of "Common Errors in the Management of Idiopathic Clubfeet Using the Ponseti Method: A Review of the Literature"

_children, 2023, doi:10.3390/children10010152_

Round 1

Reviewer 1 Report

The overall study question seems reasonable, but perhaps over-reaching. Conducting a systematic review of "complications" associated with using the Ponseti Method is way to broad a topic. Perhaps the authors should consider focusing on early treatment (casting and tenotomy), as their literature search is incomplete with regard to bracing and follow-up. 

In particular, the failure to comply with bracing has been reported in the vast majority of clubfoot outcome studies (including more than 10 that I have co-authored, none of which have been cited), yet the authors seemed to identify only 28 studies (not listed) that met the inclusion criteria, of which, only 6 are listed in Table 2. Where are the other 22? Why were so many excluded? 

Importantly, the dates of the literature search do not make sense. Ponseti did not "introduce" his method until the early 1960's and it wasn't widely adopted until the late 1990's, after his long-term results were published. Please correct this date to at least be after the publication/introduction of the Ponseti method, not 1948.

There have been numerous studies to report short-term, mid-term, and long-term outcome, and nearly all include difficulties with patient compliance. Some even plot survival outcome for brace compliance v. non-compliance... A total of 6 studies seems very low, particularly when one of the 6 appears to be a self-citation.

Author Response

Thank you for review of our manuscript and your suggestions for improving our manuscript.

1) This manuscript was submitted as a "review of the literature" and not as a "Systematic review". We understand that a systematic review would need to be done as per PRISMA Guidelines. But, we have not used these guidelines as this was a simple "review of the literature".

2) The focus of the paper is "Errors associated with the Ponseti method". 

3) We included studies which reported errors encountered with the "Ponseti method". Thus, we excluded articles that did not report on "Errors associated with the treatment of idiopathic clubfoot using the Ponseti Method".

4) We agree that Dr. Ignacio Ponseti first developed his method of treatment in 1948 and published his first outcomes study in 1963. This was followed by other outcome studies by his group from Iowa. He published his book on this method in 1996. The methods section has been changed to reflect these dates.

5) Regarding reported outcomes of the Ponseti method  for treatment of idiopathic clubfoot, we agree that there are a number of papers reporting from all over the world and we have included 2 references to systematic reviews. 

a) Gelfer, Y., S. Wientroub, K. Hughes, A. Fontalis, and D M Eastwood. "Congenital Talipes Equinovarus: A Systematic Review of Relapse as a Primary Outcome of the Ponseti Method." The Bone & Joint Journal101-B.6 (2019): 639-45. Web. This includes 84 studies of outcomes of The Ponseti method both at short and long term with relapse as the primary outcome.

b) Rastogi A, Agarwal A. Long-term outcomes of the Ponseti method for treatment of clubfoot: a systematic review. Int Orthop. 2021 Oct;45(10):2599-2608. This includes 14 studies with outcomes reported at 10 or more years.

Reviewer 2 Report

The manuscript is a review and aimed to update clinicians and provide broad guidelines that can be followed to avoid and manage these errors to optimize short and long-term outcomes of treatment of idiopathic clubfeet using the Ponseti method. A literature search was performed using the following keywords: “Idiopathic Clubfoot” (All Fields) AND “Management” OR “Outcomes” (All Fields). Databases searched included PubMed, EMBASE, Cochrane Library, Google Scholar and SCOPUS (age range: 0-12 months). A full text review of these articles was then performed looking for “complications” or “errors” reported during the treatment process.

I read the article with interest, the title is well thought out and faithfully reflects the content of the study. 

The abstract is very useful to frame the purpose of the study. 

In the introductionthe characteristics of Clubfoot have been describedThe materials and methods have been adequately developmentThe discussion is sufficiently described.

Nevertheless, some minor changes are needed to be considered suitable for publication. 

Comment 1: In the introduction: Some information about etiology, diagnosis and treatment of clubfoot should be deepeneplease adding appropriate bibliographical references(Alberghina F. et al (2021) " From Codivilla to Ponseti: historical narrative review on clubfoot treatment in Italy").

Comment 2In the materials and methodplease clearly indicate the inclusion and exclusion criteria of the centers in this study.

Comment 3: In the materials and method: the subdivision into tables as well as use of Preferred Reporting Items for Systematic Reviews and Meta-Analyzes (PRISMA) is very adequate to clarify the study results.

Comment 4: In the materials and method: The characteristics of the study should be clarified. Comment 5: In the discussion: It would be better referring to previous studies performed on the same topic, for example (Pavone V. et al (2021) "Sport Ability during Walking Age in Clubfoot-Affected Children after PonsetiMethod: A Case-Series Study ").

Comment 6In the discussionIt would be advisable to clearly refer to the limitations of the study.

Comment 7: In the discussion: For completeness I would add important data about developmental milestones in patients with idiopathic clubfoot treated by Ponseti method: (Pavone et al (2022) "Early developmental milestones in patients with idiopathic clubfoot treated by Ponseti method").

Comment 7Finally, additional English editing is needed. The Non-Native Speakers of English Editing Certificate was not signed.

Author Response

Thank you for your suggestions to improve the manuscript. Please see below our detailed response to your questions.

1) We have modified the introduction by providing more detail on the etiology of idiopathic clubfoot and the Ponseti treatment protocol. We have added references 62 and 63 to address this. Due to limitations on the word count, we have not discussed the details. We have added following sentences to the introduction: 

a) "Recent evidence suggests that the etiology of idiopathic clubfoot is multifactorial including both genetic and environmental factors." 

b) The casting phase consists of gentle manipulation followed by weekly serial stretch casting of the clubfoot using long leg Ponseti casts to correct midfoot cavus, forefoot adduction, hindfoot varus and ankle equinus (CAVE) in that specific order. This is followed by a percutaneous tendoachilles tenotomy in most cases (about 90%) to achieve ankle dorsiflexion of more than 10 degrees and a final long leg Ponseti cast with the foot abducted to about 60 degrees for 3 weeks. The bracing phase consists of using bilateral boots and a foot abduction bar full time for 3 months and part time (night and nap time) for up to 4-5 years of age, to maintain correction and prevent relapse of the clubfoot deformity.

2) We have included published studies done at various centers and have not restricted to certain centers as it is a review of literature. 

3) Since it is a literature review and not a systematic review, we have not used PRISMA guidelines.

4) Since we have conducted a literature review, we have included all studies including case series. We have modified the materials and methods section to clarify the inclusion and exclusion criteria.

5) We have not included this information as we have not looked into sports ability in walking age children after the Ponseti method. This study was focused on the "Errors associated with the Ponseti method".

6) Our study has several limitations. It is a review of literature and there are limited randomized controlled trails on the subject. There is heterogeneity of patient/physician reported outcomes. We acknowledge that this may result in inaccurate reporting of errors. Also, there is no data on level of competency/experience of the healthcare personnel performing the Ponseti casting. We do not know what is the optimum experience to minimize errors. There is a need for universal error/adverse outcome reporting system while treating clubfeet. We have added this information to the manuscript in the discussion section (last paragraph) on page 17.

7) The role of neurodevelopmental challenges that children with clubfeet face and it's impact on the Ponseti method and brace tolerance was addressed on page 16 and reference 47 was used. The recommended reference does not show any correlation between Early developmental milestones and errors associated with the Ponseti method and hence was not added to the bibliography.

8) We have got the manuscript reviewed by native English-speaking expert.

Reviewer 3 Report

Dear Author,

Thank you for the opportunity to review this article.

I congratulate you on the initiative to illustrate treatment errors in Clubfoot. Indeed, we also found that Ponseti Method has a long learning curve and the aggressivity of the treatment must target a „sweet spot” between the risk of relapse and the risk of complications like tibial fractures and skewfoot, vertical talus, and flatfoot. Unlike the literature, we prefer to treat the children in FAB for a longer period associated with daytime orthopedic boots. Even so, relapse occurs sometimes and additional orthopedic and surgical treatment is needed. The follow-up needs to be extended until the end of growth.

We encourage you to mention about flatfeet when describing potential complications as it certainly occurs in some cases. Here is an interesting article about the treatment of such complication to be used as a potential reference: The Role of Arthroereisis in Improving Sports Performance, Foot Aesthetics and Quality of Life in Children and Adolescents with Flexible Flatfoot, published in Children, DOI 10.3390/children9070973.

You could also mention risk factors in the introduction for clubfoot if your study pool contains this information. Regarding prenatal risk factors for orthopedic conditions, there were several studies such as this one that point out the intrautero position of the child and the ocurence of orthopedic conditions: Obstetric fractures in cesarean delivery and risk factors as evaluated by pediatric surgeons, published in Int Orthop, DOI 10.1007/s00264-022-05547-2.

Finally, in improving the patient-doctor relationship (because you found it to be a risk factor of non-compliance) one could use 3D modeling and printing of the patients’ feet or a standard foot model to simulate the treatment iterations and anatomic dynamics in front of the parents. We recommend you this manuscript to read and cite: The use of 3D printing in improving the patient-doctor relationship and malpractice prevention, published in RJML, DOI 10.4323/rjlm.2017.279

This is a very resourceful paper and has our support for publication after a minor revision.

Author Response

Thank you for reviewing our original manuscript and for your suggestions to improve the manuscript.

1) Overcorrected clubfeet resulting in a flatfoot is a recognized complication following treatment of the idiopathic clubfoot in both the affected foot as well as the contralateral foot (secondary to bracing). The following 2 references have been included in the manuscript to allude to this.

Dussa CU, Böhm H, Döderlein L, Forst R, Fujak A. Does an overcorrected clubfoot caused by surgery or by the Ponseti method behave differently? Gait Posture. 2020 Mar;77:308-314.

Shirai Y, Wakabayashi K, Wada I, Tsuboi Y, Ha M, Otsuka T. Flatfoot in the contralateral foot in patients with unilateral idiopathic clubfoot treated using the foot abduction brace. Medicine (Baltimore). 2017 Sep;96(35):e7937

2) Risk factors for the development of clubfoot - We are discussing errors associated with the Ponseti method for treatment of idiopathic clubfeet and not the etiology of clubfoot. We therefore feel including this will not add greatly to the message of the paper. But, we have included some details about the etiology of idiopathic clubfoot in the introduction.

3) While we agree that 3D printed models may help to improve physician-patient communication. However, these are expensive and facilities to develop these models may not be easily available in Low middle income countries. Also, these may help while developing patient specific treatment plan for more complex pediatric orthopedic conditions like spine deformities, deformities of the upper and lower extremity.  

The following reference has been included in the manuscript in the discussion section on page 17.

Traynor G, Shearn AI, Milano EG, Ordonez MV, Velasco Forte MN, Caputo M, Schievano S, Mustard H, Wray J, Biglino G. The use of 3D-printed models in patient communication: a scoping review. J 3D Print Med. 2022 Mar;6(1):13-23. 

Thank you,

Mohan

Reviewer 4 Report

This is a very interesting review on a frequently diagnosed pathology all around the world. However, this article should clarify some issues.

First of all, please mention what was the process of using the images without having a consent from an ethical committee or from in-laws? You said that ethical committee approval was waived due to not using humans, but all the figures imply fotos of human beings and children.

Secondly, in Figure No 1 - there is a face that can be distinguished from the image - this should be hidden.

The Results section should be clearly more in depth. While the two tables provide value, it is not sufficient.

English, spelling and grammar are ok.

Author Response

Thank you for review of our manuscript and your suggestions for improving our manuscript.

1) This was a secondary research study and did not involve getting data directly from patients  and hence was IRB exempt. However, we got consent from patients' and their legally authorized guardians to use clinical photographs to highlight some complications associated with the Ponseti method.

2) In Figure no. 1, the face has now been completely hidden in the resubmission.

3) Results section: This is a secondary study and hence we do not have any more data to provide than what has been provided in the results section.

4) The software "Grammarly" was used to check spelling and grammar. We also used the "Spelling & Grammar" function in Microsoft word to complete the same.

Thank you,

Round 2

Reviewer 1 Report

The authors have still failed to do a comprehensive review of the literature. For example, there have been entire reviews dedicated to errors associated with failure to strictly follow Ponseti's method, including but not limited to: choice of cast material, duration of bracing, how to treat a relapsed deformity, whether or not to perform a tenotomy, etc. This is a huge topic. My suggestion remains to focus this manuscript on a narrower topic, as technically, there are hundreds of manuscripts that have reported "errors" or adverse outcomes, based on failure to actually follow Ponseti's method.

Author Response

Thank you for your comments.

We would like the reviewer to provide examples of systematic reviews done on errors related to the Ponseti method. 

In our review of the literature, we came across systematic reviews on the Ponseti method, outcomes related to relapses but not specifically related to errors with the Ponseti method. 

Thank you

Mohan

Reviewer 4 Report

Authors have made the changes. 

Author Response

Thank you for your suggestions.

We have made all changes as recommended by the reviewer.